# Prompting Large Language Models with Chain-of-Thought for Few-Shot Knowledge Base Question Generation

**Yuanyuan Liang[1], Jianing Wang[1], Hanlun Zhu[1], Lei Wang[2]**
**Weining Qian[1], Yunshi Lan[1*]**
[1] East China Normal University, [2] Singapore Management University
leonyuany@stu.ecnu.edu.cn, {lygwjn, timberflowing}@gmail.com
lei.wang.2019@phdcs.smu.edu.sg, {wnqian, yslan}@dase.ecnu.edu.cn

## Abstract

The task of Question Generation over Knowledge Bases (KBQG) aims to convert a logical form into a natural language question. For the sake of expensive cost of large-scale question annotation, the methods of KBQG under low-resource scenarios urgently need to be developed. However, current methods heavily rely on annotated data for fine-tuning, which is not well-suited for few-shot question generation. The emergence of Large Language Models (LLMs) has shown their impressive generalization ability in few-shot tasks. Inspired by Chain-of-Thought (CoT) prompting, which is an in-context learning strategy for reasoning, we formulate KBQG task as a reasoning problem, where the generation of a complete question is split into a series of sub-question generation. Our proposed prompting method KQG-CoT first selects supportive logical forms from the unlabeled data pool taking account of the characteristics of the logical form. Then, we construct a task-specific prompt to guide LLMs to generate complicated questions based on selective logic forms. To further ensure prompt quality, we extend KQG-CoT into KQG-CoT+ via sorting the logical forms by their complexity. We conduct extensive experiments over three public KBQG datasets. The results demonstrate that our prompting method consistently outperforms other prompting baselines on the evaluated datasets. Remarkably, our KQG-CoT+ method could surpass existing few-shot SoTA results of the PathQuestions dataset by 18.25, 10.72, and 10.18 absolute points on BLEU-4, METEOR, and ROUGE-L, respectively.

## 1 Introduction

Question generation task requires a system to produce natural language questions based on the given context. KBQG (Guo et al., 2022) is one of the imperative question generation tasks when the given

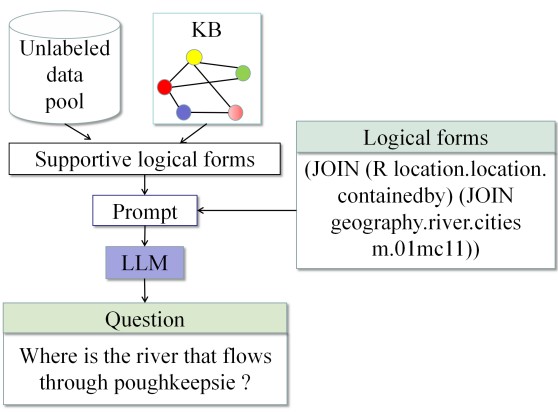

Figure 1: Overview of KQG-CoT framework.

context derived from Knowledge Bases (KBs) is in the form of logical. KBQG has attracted increasing interests from both the industry and academia due to its potential for data augmentation in QA systems (Xiong et al., 2022; Chen et al., 2023) and its ability to assist dialogue systems in creating coherent questions (Lee et al., 2018).

Existing studies (Kumar et al., 2019; Ke et al., 2021; Fei et al., 2022; Guo et al., 2022; Chen et al., 2023) for KBQG tasks have predominantly utilized neural network-based approaches and demonstrated impressive performance by conducting fine-tuning on extensive training datasets. However, as the collection of KBQG data is labor-intensive, researchers start paying attention to the few-shot KBQG tasks (Xiong et al., 2022), where a great challenge is posed for suppliers with limited resources: 1) A great deal of annotated data is demanded to allow the existing fine-tuned models to generalize well over different logical forms. However, due to the limitations of low-resource availability, training conventional models by fine-tuning on the full data becomes unrealistic. 2) A logical form is composed of entities, relations, and query grammar. Having logical forms with various combinations of these basic components is crucial to

---

*Corresponding author

uphold the model's capability for compositional generalization. The lack of data leads to a compositional challenge to the KBQG tasks (Gu et al., 2021). 3) Certain logical forms can become complex when operations such as aggregation, superlatives, and comparisons are involved. Representing these logical forms presents additional challenges. Moreover, developing a KBQG method that incorporates diverse and elaborate expressions becomes particularly difficult in such low-resource scenarios (Xiong et al., 2022; Guo et al., 2022).

Recently, LLMs such as GPT-3 and Codex (Gao et al., 2022; Suzgun et al., 2022; Wei et al., 2022; Wang et al., 2023a) have proven their strong generalizability on a wide range of few-shot and zero-shot tasks with CoT, including text interpretation, computer vision, planning and reasoning. Meanwhile, a line of work (Kasner et al., 2022; Moiseev et al., 2022; Andrus et al., 2022; Trajanoska et al., 2023; Xie et al., 2023) validates that LLMs have the strong capability to accurately capture the semantics of relations between values in the data, enabling to transform the structured instructions to narrative text. The above studies inspire us to explore few-shot KBQG tasks by prompting LLMs with CoT.

However, how to apply LLMs to KBQG with CoT is still unclear. On one hand, KBQG differs from tasks like code generation or question answering, as it involves incorporating KB-specific items into the input instead of self-contained narratives. Therefore, formatting the input in an easily understandable manner while considering the KB schema is crucial. On the other hand, the challenge lies in designing effective CoT prompts (Wei et al., 2022) that can enhance the performance of LLMs in the context of few-shot KBQG.

In this work, we propose KQG-CoT framework, which is the first attempt for training-free few-shot KBQG with LLMs. As shown in Figure 1, our framework consists of two main steps, the objects of which are supportive logical forms selection from an unlabeled data pool and prompt construction. To acquire coherent logical forms, we employ a clustering technique to carefully choose multiple logical forms that serve as representatives, considering both their syntactic and semantic characteristics. To construct prompt, inspired by the principle of CoT (Wei et al., 2022), we take the selected logical forms as exemplars and write rationales to split the generation of a complete question into mul-

tiple steps. We concatenate the above rationales with the queried logical form to form a prompt, which guides a LLM to outcome a reasoning process of generating a complex question aligning with the logical form. We further improve KQG-CoT to KQG-CoT+ via sorting the supportive logical forms by complexity.

As previous methods rely heavily on the training instances to fine-tune a KBQG model. KQG-CoT does not need numerous logical form question pairs to train the models. We test the performance of our prompting methods under few-shot setting on three public datasets, namely WebQuestions (Kumar et al., 2019), PathQuestions (Zhou et al., 2018), and GrailQA (Gu et al., 2021). We conduct a comprehensive comparison with a range of commonly used CoT baseline methods including Auto-CoT (Zhang et al., 2023c), Active-CoT (Diao et al., 2023), Random-CoT (Brown et al., 2020) and so on. The experimental results show that we can outperform all of them with an observable margin. Besides, we also compare with a set of SoTA systems trained with full data or few data. Our few-shot method could achieve competitive results to the full training methods. Remarkably, our few-shot method could surpass existing few-shot SoTA results of PathQuestions dataset by 18.25, 10.72 and 10.18 absolute points on BLEU-4, METEOR and ROUGE-L, respectively.

KQG-CoT provides a simple but effective solution to few-shot KBQG problem, we expect it could serve as an important baseline for future investigation to KBQG tasks under low-resource scenarios.

Our main contributions are summarized as follows:

- By encoding and clustering the skeletons of logical forms, we successfully retrieved supportive logical forms that are particularly suitable for constructing effective prompts.

- We reorganized the sequence of examples and utilized the CoT method to construct prompts that are highly effective for large language models.

- The experimental results indicate that our method surpasses the baseline by a significant margin and achieves performance levels that are comparable to fine-tuned methods.

## 2 Related Work

**Knowledge Base Question Generation.** The early approaches for KBQG tasks are template-based methods. Berant et al. (2013 and Talmor and Berant (2018a) utilized search engines and manual annotation to construct the natural language questions based on logical forms. However, template-based methods rely on manual intervention, which is hard to be scaled up. With the advancement of deep neural networks, neural network-based methods have emerged as a prominent and widely adopted approach. Kumar et al. (2019) and Chen et al. (2023) proposed end-to-end models based on Transformer and Graph2seq models, which are capable of generating complex, multi-hop questions based on a subgraph. Follow-up studies (Fei et al., 2022; Guo et al., 2022) developed more complicated models for KBQG, which ensure the relevance between the generated questions and subgraphs. Xiong et al. (2022) proposed a method for low-resource KBQG, where an auto-prompter is developed to paraphrase a logical form into a description, so that a pre-trained language model can be fine-tuned with the augmented data. Our work is different from this one as our method focuses on solving few-shot KBQG challenge with frozen LLMs.

**Few-shot Learning for Text Generation.** In recent years, significant progress has been made in the field of few-shot learning for text generation. One line of work develops meta-learning frameworks for text generation (Mi et al., 2019; Madotto et al., 2019; Zeng et al., 2021; Hospedales et al., 2022), which aims to acquire an optimal initialization that enables accurate and rapid adaptation to a new task, even when limited data is available. Other line of work proposes different augmentation algorithms to synthesize the data for training (Song et al., 2019; Zhao et al., 2022), so that conventional text generation models can be applied to the augmented data. Most recently, LLMs are leveraged to solve few-shot text generation tasks such as text summarization (Yang et al., 2023; Zhang et al., 2023b; Liu et al., 2023), machine translation (Wang et al., 2023b; Hendy et al., 2023), dialogue generation (Zhang et al., 2023a; Valvoda et al., 2022; Kang et al., 2022) and so on. There is no existing study applying LLMs to few-shot KBQG tasks.

**In-Context Learning with LLMs.** Without gradient updates, In-Context Learning (ICL) effectively tackles a wide range of NLP tasks by incorporating a small number of prompted examples as part of the input (Ruis et al., 2023) to help LLMs understand the tasks. Multiple studies (Su et al., 2022; Rubin et al., 2022) explored the selection of examples that are similar to the query during prompt construction. Recent researches (Lu et al., 2022a; Liu et al., 2022; Diao et al., 2023; Wang et al., 2023c) highlight that the order of these examples in the prompt has a substantial influence. CoT is a prompting strategy decomposing complex tasks into sub-tasks, helping the model to derive the correct answers progressively (Wei et al., 2022; Zhou et al., 2023). It has been widely used in mathematical word problem solving, common-sense reasoning, and symbolic reasoning. Our work incorporates CoT strategy into KBQG tasks, where iterative process enables LLMs to ultimately obtain a complex question aligning with the logical form.

## 3 Methodology

### 3.1 Problem Formulation

A KB consists of a set of triples. A logical form is a structural expression of a subgraph in the KB, which may consist of complex operations (e.g., aggregation, comparative and superlative) and can be utilized to execute against a KB. The task of KBQG requires a system to generate a natural language question when given a logical form and the corresponding KBs with consistent semantics.

### 3.2 Method Overview

Recently, the LLM has shown its impressive in-context few-shot learning capabilities. Instead of fine-tuning a pre-trained model to adapt it to a downstream task, we can simply apply it to a new task with a few examples as prompt during inference (Yang et al., 2022; Li et al., 2023). For the KBQG task, we adopt a two-stage method to design CoT prompts, which effectively enable the LLM to comprehend complex logical forms and generate questions. Concretely, the first stage **Supportive Logical Forms Selection** focuses on identifying supportive examples that represent various syntax patterns of logical forms. To accomplish this, we encode the structure of logical forms, perform clustering, and employ sampling techniques to select top-$k$ supportive logical forms. Once these supportive examples are selected, we leverage LLMs with CoT prompts to generate natural language questions. This leads us to the second stage, **Prompt Construction**, which involves producing sub-questions as rationales. Through this process,

we can ultimately formulate a complex question that adequately captures the semantic of the logical form. A schematic diagram of our method is displayed in Figure 2.

### 3.3 Supportive Logical Forms Selection

Zhang et al. (2023c) has shown that when constructing demonstrations, we need to mitigate the effect of few-shot CoT errors by differentiating the design of demonstrations. In KBQG tasks, supportive logical forms are those that can cover diverse logical rules, so as to offer more syntax information for LLMs to generate questions. Unlike the narrative inputs, the logical form is a combination of program structures and schema items (i.e., entities and relations). Therefore, it is essential to take both aspects into consideration when selecting supportive logical forms. In our approach, we utilize **Structure Encoding and Clustering**, followed by a **Logical Form Sampling** process to select supportive logical forms.

**Structure Encoding and Clustering**. To ensure the logical forms can be drafted for unseen questions, we extract their structures by converting the schema items into symbolic variables. Specifically, we keep the grammars in the logical form unchangeable. Then, we replace the relation with symbol "$r$" and we replace the entity with "$e$". This structure is also known as a abstract query graph (Chen et al., 2021), which reflects the topology and the component classes of logical forms. For instance, the raw logical form is:

```
(AND medicine.routed_drug
(JOIN medicine.routed_drug.marketed_formulations
m.0hqs1x)).
```

It becomes the following structure after conversion:

```
(AND r (JOIN r e)).
```

Once we have obtained the structure of the logical forms, which filters out the semantic meaning of the logical forms. We encode the structure representation into a fix-length embedding. In detail, we view the structure as a sequence of tokens. We encode the contexts of the sequence with Sentence-Transformers (Reimers and Gurevych, 2019), which is an advanced model for text embedding. The encoded vectors are well-suited for calculating the similarity between sentences. We extract the final hidden state of as the vectorized representation of the sentence. After that, we utilize

the K-means (Hartigan and Wong, 1979) clustering algorithm to group the encoded structure into $k$ clusters based on their syntactic similarity.

**Logical Form Sampling**. Each cluster contains a group of logical forms with the similar structure, we randomly pick up a structure from each group and obtain $k$ representative structures. As each structure may correspond to multiple logical forms. We further identify $k$ logical forms with distinct semantics deriving from the $k$ selected structures. To this end, we iteratively sample logical forms holding the maximum diversity of semantics. Specifically, for the first logical form, we randomly pick up one from the candidates. Then we search logical forms for another structure. We greedily pick up a candidate with least semantic similarity to the selected logical forms, where the similarity is measured by the encoding of the original logical forms. We repeat the process until we have gone through $k$ structures as shown in Figure 2.

To help the LLMs fully understand the logical forms, we substitute the entities in the original logical forms with their surface names in the KB. In this way, we obtain $k$ supportive logical forms.

### 3.4 Prompt Construction

Since some logical forms have complicated semantics and even nested syntactic structures are included. Following the CoT method, we construct a reasoning chain prompt based on the supportive logical forms retrieved above. For each example, we need to generate a reasoning chain based on logical forms to elicit LLMs generate questions from simple to complicated. To this end, we hold two criteria when constructing reasoning chains:

(i) The templates should break up the generation of a complicated question into a step by step process.

(ii) The templates should clearly identify the subcomponent in a logical form that requires LLMs to focus on for each step.

Therefore, we first break down a logical form in a nested manner, where the follow-up logical forms include the preceding logical forms. Specifically, the first step usually generates a simple question querying one-hop relation from the topic entity. The second step usually generates a question querying two-hop relation chain involving the above one-hop relation. As we can see from Figure 2, the first step of prompt

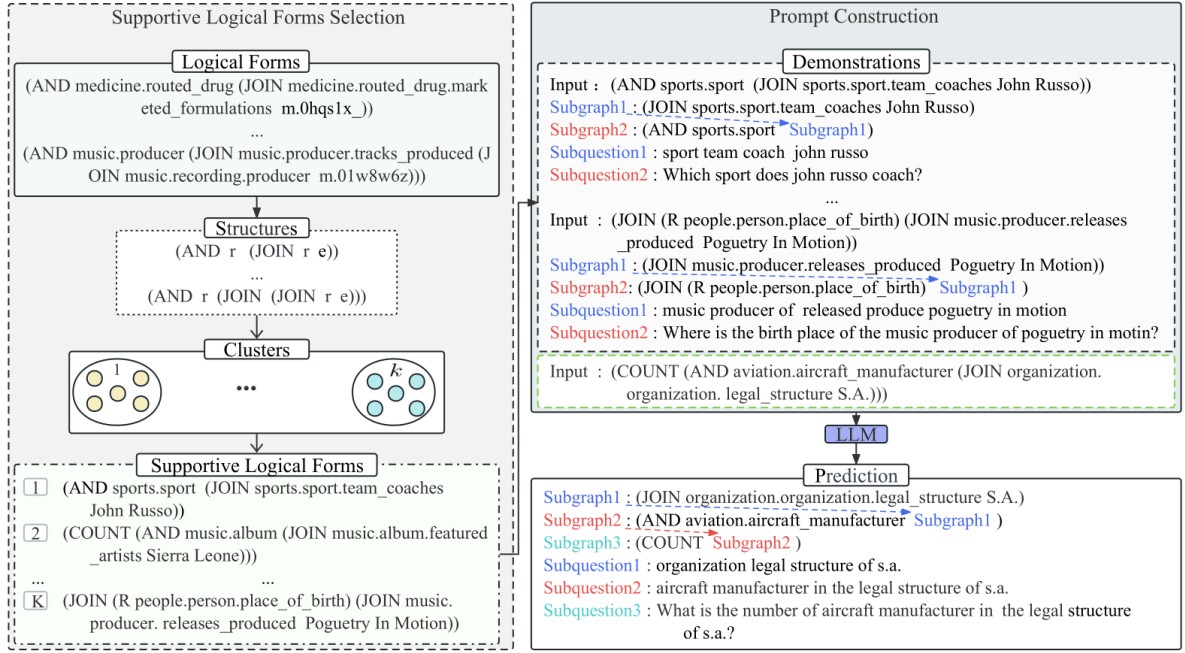

Figure 2: KQG-CoT framework. The supportive logical forms are selected from an unlabeled data pool by extracting the structures, clustering the structures and sampling the most representative ones. A total of $k$ demonstrations are automatically constructed using reasoning chains. The tested logical form is appended to the demonstrations to form the complete prompt, which can elicit the LLM to generate a series of subquestions sequentially from simple to complicated. Finally, the last subquestion can be extracted as the final prediction.

parses the entire logical form into one-hop relation *subgraph1* "(AND sports.sport.team_coaches John Russo)" which leads to a simple *subquestion1* "*sport team coach john russo* ". The second step includes the parsed logical form appended to the previous step as a component and generates question "*Which sport does john russo coach?*" based on the *subgraph2* and *subquestion1*. As a result, we continuously expand the logical form until a complete question is formed. This step-by-step process ensures that the generated question is semantically coherent and grammatically accurate.

During inference, we concatenate all the demonstrations and queried logical form as the final prompt. Based on the example in Figure 2, the prompt includes "*Input: (AND ... Input: (JOIN ... Input: (COUNT ... S.A.*". After receiving the prompt, LLMs outcome the predictions that clarifies the intermediate generation steps of *subquestion1*, *subquestion2*, and *subquestion3*. And the last subquestion will be our final predicted question, which is "*What is the number of aircraft manufacturer in the legal structure of s.a. ?*".

| Dataset | #Q | #R | #E | #T |
|---------|------|------|--------|----------|
| WQ | 22,989 | 672 | 25,703 | 2/99/5.8 |
| PQ | 9,731 | 378 | 7,250 | 2/3/2.7 |
| GQ | 64,331 | 3,720 | 32,585 | 1/4/1.4 |

Table 1: Statistics of the evaluated datasets. #Q denotes the number of questions. #R and #E denote the total number of relations and entities, respectively. #T denotes the minimum/maximum/average number of triplets involved in each question.

## 4 Experiment

In this section, we first introduce the KBQG datasets used to evaluate the performance of our proposed method and the comparable baseline methods. Next, we present the implementation details and demonstrate the experimental results.

### 4.1 Data and Metrics

We evaluate our prompting method on the following three public datasets:

**WebQuestions (WQ)** (Kumar et al., 2019)[1] is a KBQG dataset combining instances from WebQuestionsSP (Serban et al., 2016) and Com-

---

[1] https://github.com/liyuanfang/mhqg

| Method | WQ | | | PQ | | | GQ | | |
|---|---|---|---|---|---|---|---|---|---|
| | B | M | R | B | M | R | B | M | R |
| Standard Prompt | 24.86 | 29.01 | 52.74 | 55.87 | 42.24 | 76.83 | 29.17 | 33.52 | 42.95 |
| Random-CoT | 25.02 | 29.37 | 53.16 | 56.42 | 42.61 | 77.03 | 29.81 | 33.75 | 43.31 |
| Manual-CoT | 28.44 | 30.24 | 54.30 | 60.37 | 42.88 | 77.48 | 30.18 | 33.61 | 44.89 |
| Active-CoT | 26.02 | 29.55 | 54.01 | 58.78 | 43.86 | 76.78 | 30.27 | 33.71 | 44.07 |
| Auto-CoT | 28.42 | 29.65 | 53.47 | 59.59 | 43.16 | 77.13 | 30.17 | 34.22 | 44.47 |
| KQG-CoT (Ours) | 28.89 | 30.41 | 54.38 | 60.81 | 43.54 | 77.35 | 30.51 | 34.26 | 44.91 |
| KQG-CoT+ (Ours) | **29.73** | **31.08** | **55.14** | **61.71** | **44.27** | **78.41** | **31.24** | **34.94** | **45.36** |

Table 2: Few-shot evaluation of existing prompting methods with Frozen LLMs on three KBQG datasets. The best and second best results are boldfaced and underlined respectively.

plexWebQuestions (Talmor and Berant, 2018b). It provides questions, answers, and annotated subgraphs. This dataset is commonly evaluated in existing work (Guo et al., 2022).

**PathQuestions (PQ)** (Zhou et al., 2018)[2] is a commonly used KBQG dataset constructed from a KBQA dataset. It contains questions inquiring a chain of relations, wherein the path between the topic entities and answer entities is 2-hop or 3-hop.

**GrailQA (GQ)** (Gu et al., 2021)[3] is a large-scale KBQA dataset built on Freebase, which covers 86 domains. It covers complex questions which require counting, ranking and even superlative inquiry. Each question is associated with a s-expression, which can be viewed as a logic form.

We collect the annotated the logic form from the training set as the data pool and leave the original questions untouched. The questions in the validation or test set are sampled to evaluate our method. Statistics of evaluated datasets are shown in Table 1.

Following previous KBQG studies, we rely on a set of well-established metrics as for KBQG evaluation: BLEU-4 (Papineni et al., 2002), METEOR (Banerjee and Lavie, 2005) and ROUGE-L (Lin, 2004). BLEU-4 and ROUGE-L can be viewed as precision and recall for text generation tasks, respectively. METEOR is a comprehensive metric beyond exact matches, which also accounts for partial matches and variations in word order. We denote them as **B**, **M** and **R**, respectively.

### 4.2 Comparable Methods

We denote our prompting method as **KQG-CoT**. Previous studies (Lu et al., 2022b) have proven that the order of the exemplars is significant to the prompt results, we implement an improved version

by sorting the demonstrations from short to long after sampling. We denote this method as **KQG-CoT+**.

As there is no existing attempt for few-shot KBQG tasks with LLMs, we adopt five general prompting methods under few-shot scenarios as our baselines.

**Standard Prompt** (Brown et al., 2020) is a standard prompting method of in-context learning, where $k$ random logical forms and questions are concatenated to form the prompt. The prediction is one-step generation.

**Random-CoT** is an intuitive CoT prompting baseline where $k$ logical forms are randomly selected from the data pool and we follow the original work (Brown et al., 2020) to describe the sub-task in a narrative.

**Manual-CoT** (Wei et al., 2022) is a CoT prompting with $k$ human-written exemplars as demonstrations and the sub-task is presented in narratives.

**Active-CoT** (Diao et al., 2023) is an ensemble framework for CoT prompting. The multiple logical forms are randomly selected as a validation set. Then multiple measurements (e.g., disagreement, variance) are leveraged as the uncertainty value for each logical form to produce the final question.

**Auto-CoT** (Zhang et al., 2023c) automatically constructs prompt by selecting $k$ demonstrations with a cluster-based algorithm and the sub-task is presented in narratives. We simply adopt the prompting method to KBQG tasks by encoding all logical form in a textual way.

### 4.3 Implementation Details

For encoding of logical forms, we utilize all-MiniLM-L6-v2[4] checkpoint from the Sentence-Transformers library in Huggingface for effective

---

[2]https://github.com/zmtkeke/IRN
[3]https://dki-lab.github.io/GrailQA/

[4]https://huggingface.co/sentence-transformers/all-MiniLM-L6-v2

encoding. As this is a few-shot scenario, we manually write the rationales for the $k$ demonstrations in the chain prompt. We utilize `text-davinci-003` from OpenAI API[5] to generate questions and set the number of clusters as $k = 12$[6].

### 4.4 Main Results

| Method | WQ | | |
|---|---|---|---|
| | B | M | R |
| *Full Training* | | | |
| L2A (Du et al., 2017) | 6.01 | 26.95 | 25.24 |
| Transformer (Vaswani et al., 2017) | 8.94 | 13.79 | 32.63 |
| MHQG (Kumar et al., 2019) | 11.57 | 29.69 | 35.53 |
| BiGraph2Seq (Chen et al., 2023) | 29.45 | 30.96 | 55.45 |
| T5-Large (Raffel et al., 2020) | 28.78 | 30.55 | 55.12 |
| JointGT (Ke et al., 2021) | 30.02 | 32.05 | 55.60 |
| IGND (Fei et al., 2021) | 30.62 | 31.41 | 55.82 |
| LFKQG (Fei et al., 2022) | **31.66** | **32.69** | 56.75 |
| DSM (Guo et al., 2022) | 28.62 | - | **64.25** |
| *Few-shot Evaluation* | | | |
| KQG-CoT | 28.89 | 30.41 | 54.87 |
| KQG-CoT+ | 29.73 | 31.08 | 55.46 |

Table 3: Comparison between few-shot evaluation of KQG-CoT/KQG-CoT+ and full-trained evaluation of other systems on WQ.

| Method | PQ | | |
|---|---|---|---|
| | B | M | R |
| *Full Training* | | | |
| L2A (Du et al., 2017) | 17.00 | 50.38 | 19.72 |
| Transformer (Vaswani et al., 2017) | 56.43 | 43.45 | 73.64 |
| MHQG (Kumar et al., 2019) | 25.99 | 33.16 | 58.94 |
| BiGraph2Seq (Chen et al., 2023) | 61.48 | 44.57 | 77.72 |
| AutoQGS (Xiong et al., 2022) | 65.13 | 47.50 | 76.80 |
| T5-Large (Raffel et al., 2020) | 58.95 | 44.72 | 76.58 |
| IGND (Fei et al., 2021) | 61.69 | 45.11 | 77.28 |
| LFKQG (Fei et al., 2022) | 63.92 | 46.91 | 78.40 |
| JointGT (Ke et al., 2021) | **65.89** | **48.25** | 78.87 |
| DSM (Guo et al., 2022) | 61.03 | - | **86.06** |
| *Few-shot Evaluation* | | | |
| BiGraph2Seq (Chen et al., 2023) | 1.01 | 4.99 | 12.07 |
| JointGT (Ke et al., 2021) | 43.15 | 35.91 | 69.57 |
| AutoQGS (Xiong et al., 2022) | 43.46 | 33.55 | 68.23 |
| KQG-CoT | 60.81 | 43.54 | 77.35 |
| KQG-CoT+ | 61.71 | 44.27 | 78.41 |

Table 4: Comparison between few-shot evaluation of KQG-CoT/KQG-CoT+ and few-shot/full-trained evaluation of other systems on PQ.

**Comparison with Baselines.** Table 2 showcases the experimental results of our methods and baseline approaches. We have the following observations based on it:

1) Comparing all CoT prompting methods, in the few-shot setting, our KQG-CoT+ prompting

[5] https://openai.com/blog/openai-codex/
[6] Detailed prompt design of KQG-CoT+ is presented in Appendix A.3.

consistently outperforms other method across all KBQG datasets by a remarkable margin. Specifically, KQG-CoT+ improves the performance of the competitive Auto-CoT by 0.72 to 2.12 absolute values for all datasets. Meanwhile, KQG-CoT also outperforms existing CoT prompting methods on BLEU-4 of all the datasets.

2) Comparing CoT methods with standard prompting, we notice that all the CoT prompting methods outperform the standard prompting method, which indicates that, to generate questions with complex logic and long dependency, splitting the entire generation task into sub-tasks are crucial for maintaining the coherence and accuracy of the questions.

3) Comparing Auto-CoT, KQG-CoT and KQG-CoT+, even though all these methods adapt clustering to select $k$ demonstrations, KQG-CoT and KQG-CoT+ are more effective as we elaborately design encoding algorithm and prompt templates for KBQG tasks, which makes it fit more into the question generation from the logical forms.

**Comparison with Other Systems.** We further compare our prompting methods with other KBQG systems on the WQ and PQ datasets. According to our knowledge, we are the first to work on the KBQG task using the GQ dataset, so there are no existing methods available for comparison.

In Table 3, we can see that with 12 demonstrations, our method can outperform majority of full-trained systems on WQ dataset, where all training data is leveraged to train a model. KQG-CoT+ prompting method can achieve 29.73%, 31.08% and 55.46% for BLEU-4, ROUGE-L and METEOR respectively, which are close to the SoTA results.

In Table 4, we can see that for PQ dataset, our method can still achieve better results than most of existing full-trained KBQG models. Compared with existing methods under few-shot settings, our methods can significantly improve the BLEU-4 over AutoQGS by around 20 absolute points. It is worth noting that AutoQGS takes 0.1% training instances for training and we simply leverage 12 instances for inference, which highlights superiority of our methods.

### 4.5 More Analysis

**Human Evaluation.** We further conduct human evaluation by randomly sampling 300 examples from the test set of WQ dataset. The generated

| | | | |
|---|---|---|---|
| **Input**: (AND military.military_conflict (JOIN military.military_conflict.force_strengths (JOIN (R military.military_resource.conflicts) Bendix AN/FPS-20))) | | | |
| **Manual-CoT**: Which military conflict involves the bendix an/fps-20 and what are its force strengths? | | | |
| **Active-CoT**: What military conflict has force strengths using bendix an/fps-20? | | | |
| **Auto-CoT**: What are the force strengths in the bendix an/fps-20 military conflict? | | | |
| **KQG-COT+**: Which military conflict has force strengths with bendix an/fps-20? | | | |
| **Ground Truth**: Which military conflict has force strengths with conflicts bendix an/fps-20? | | | |
| **Input**:(AND measurement_unit.measurement_system (JOIN measurement_unit.measurement_system. heat_capacity_units Joule per kelvin)) | | | |
| **Manual-CoT**: What is the measurement system that uses joules per kelvin for heat capacity units? | | | |
| **Active-CoT**: What is the measurement system for heat capacity units of joule per kelvin? | | | |
| **Auto-CoT**: Which measurement system uses joule per kelvin as its heat capacity unit? | | | |
| **KQG-COT+**: What measurement system uses joule per kelvin as a units to measure heat capacity? | | | |
| **Ground Truth**: What system uses joule per kelvin as the unit to measure heat capacity? | | | |

Table 5: Illustrative examples from KQG-CoT+ and baseline methods on GQ.

| Model | Synt. | Comp. | Relev. |
|---|---|---|---|
| Ground Truth | 4.88 | 4.92 | 4.91 |
| Standard Prompt | 3.67 | 3.76 | 3.99 |
| Random-CoT | 4.05 | 4.21 | 4.12 |
| Manual-CoT | 4.60 | 4.54 | 4.72 |
| Active-CoT | 4.56 | 4.71 | 4.75 |
| Auto-CoT | 4.38 | 4.77 | 4.55 |
| KQG-CoT+ | **4.63** | **4.80** | **4.78** |

Table 6: Results of human evaluations on WQ. Synt., Comp. and Relev. denote syntactic correctness, complexity and relevance, respectively.

| Method | GQ | | |
|---|---|---|---|
| | B | M | R |
| KQG-CoT+ | 31.24 | 34.94 | 45.36 |
| (a) w/o CoT | 30.11 | 33.58 | 43.88 |
| (b) K-means → Random | 29.81 | 33.75 | 43.31 |
| (c) w/o structure encoding | 30.03 | 33.41 | 43.76 |

Table 7: Ablation study of our KQG-CoT+ method on GQ.

questions are rated on a scale of 1 to 5 considering the aspects of syntactic correctness, complexity, and relevance to the given logical forms. We ask three annotators to score the generated questions with 1-point being poor and 5-point being perfect. The score of each question is averaged over all annotators. We present the results in Table 6, where we can observe a similar trend between human and automatic evaluation. Our approach outperforms all comparable methods, the evaluated scores of which are close to the ground truth.

**Ablation Study.** We conduct ablation study to assess the effectiveness of components of our model and display the results in Table 7. We first exclude the CoT reasoning chain, and observe a performance drop of the evaluate metrics. This indicates

that CoT plays an important role in generating complicated questions. Then we remove the K-means algorithm and randomly select supportive logical forms. The decrease of the results indicates that our clustering algorithm could provide more diverse logical forms as our demonstrations. We further encode the entire logical forms without extracting their structures. The results decrease which indicate that the structure is a significant indicator to obtain the clusters[7].

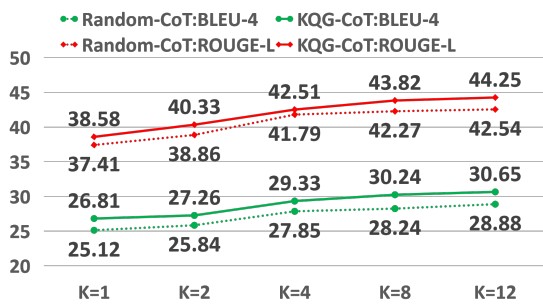

Figure 3: The BLEU-4 and ROUGE-L scores of our method and Random-CoT with increasing number of shots on GQ.

**Effect of $k$.** We investigate the effect of $k$ in Figure 3. As observed, with an increase of the number of demonstrations, both our methods and Random-CoT show increasing BLEU-4 and ROUGE-L scores. This indicates that the number of demonstrations is significant in activate the potentials of LLMs. Compared with Random-CoT, our method shows a larger gain when the value of $k$ becomes large, this indicates that our methods indeed pick up the most representative logical form as the demonstrations.

**Case Study.** To provide a comprehensive compar-

---
[7]The ablation study on WQ and PQ is presented in Appendix A.1.

ison between KQG-CoT+ method and the baseline models on GQ dataset, we present multiple example cases in Table 5. Our method elicits the intermediate generation steps and provides more guidance to LLMs so that our KQG-CoT+ generates questions that are grammatically correct and semantically close to the given logical form. In contrast, baseline methods may encounter issues such as inconsistency in the logical form, misplaced modifiers, or unsmooth expressions.

**Effectiveness of Structured Encoding and Clustering.** To demonstrate the effectiveness of the proposed Structured Encoding and Clustering in selecting diverse structures, we conducted a quantitative assessment of the average semantic similarity between the logical forms extracted using our method and the baseline method at K=8 on the GrailQA dataset. The results are presented in Table 8. The data from the initial segment, shown in the table below, reveals that the logical forms chosen by our method exhibit a lower average semantic similarity. When viewed collectively, these findings offer strong evidence for the efficacy of our proposed approach.

| Method | Average_similarity |
|---|---|
| Random | 0.285 |
| Active-CoT | 0.274 |
| Auto-CoT | 0.265 |
| KQG-CoT | 0.252 |

Table 8: The average semantic similarity between the logical forms of different methods.

**Impact of Sorted Order.** To assess the impact of the sorted order of demonstrations in KQG-CoT+, we compared the performance of Auto-CoT and Active-CoT using the same sorted order of demonstrations in KQG-CoT+ (i.e., Auto-CoT+ and Active-CoT+) and conducted experiments on the GrailQA dataset . The Table 9 shows that, compared to the Active-CoT+ and Auto-CoT+ methods, our proposed KQG-CoT+ method still exhibits significant improvements.

| Method | B | M | R |
|---|---|---|---|
| Active-CoT+ | 30.40 | 34.04 | 44.22 |
| Auto-CoT+ | 30.52 | 34.59 | 44.77 |
| KQG-CoT+ | 31.24 | 34.94 | 45.36 |

Table 9: The result data for Auto-CoT+, Active-CoT+, and KQG-CoT+ on the GrailQA dataset.

**KQG-CoT Improve KBQA Task.** To confirm the efficacy of our approach in enhancing the performance of KBQA methods, we initiated a data augmentation procedure for the WebQuestions dataset. It's important to highlight that the augmented dataset was merely half the size of the original dataset. Next, we trained the KBQA method RnG-KBQA (Ye et al., 2022) by combining the augmented and original datasets, resulting in the improved version called RnG-KBQA+. The results, as outlined in Table 10, demonstrate that we conducted a relatively straightforward augmentation on a limited dataset subset. Nevertheless, the F1 score of the original KBQA method witnessed a notable increase of 2.8%. This demonstrates that our proposed KBQG method provides significant assistance to downstream KBQA tasks[8].

| Method | F1-Score |
|---|---|
| RnG-KBQA | 75.6 |
| RnG-KBQA+ | 78.4 |

Table 10: The result of our approach in improving the performance of KBQA methods.

## 5 Conclusion

In this paper, we presented the KQG-CoT approach to tackle few-shot KBQG tasks. KQG-CoT retrieves relevant logical forms from unlabeled data and incorporates their characteristics. It then generates explicit prompt to showcase the reasoning process for complex question generation based on the selected examples. Experimental results demonstrate that our approach achieves state-of-the-art performance compared to baselines and even shows competitive results to full-training methods.

## Limitations

Our proposed prompting method, KQG-CoT, partially relies on handcrafted prompts when writing the subquestions. However, handcrafted prompts are usually based on the personal knowledge and experience of the exports, which can introduce subjective biases.

## Acknowledgements

This work was supported by Natural Science Foundation of China (Project No. 62206097) and Shanghai Pujiang Talent Program (Project No. 22PJ1403000). We sincerely thank the anonymous reviewers for their valuable comments and feedback.

---

[8]Further analysis will be presented in Appendix A.4.

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

# A Appendix

## A.1 Ablation Study on More Datasets

We display Table 12 to show more ablation studies on WQ and PQ datasets. We can also recognize the significance of our CoT reasoning chain, K-means algorithm, and structure encoding.

## A.2 Illustrative Examples of KQG-CoT+ Prompt

We present a selection of illustrative examples showcasing our proposed prompts and predictions on WQ, GQ, and PQ in Table 13, Table 14 and Table 15, respectively. As

## A.3 Detailed Prompt Design of KQG-CoT+

To enhance the guidance provided to LLM in question generation, we have included a descriptive sentence in the demonstrations, which states: "*Let's engage in a step-by-step exercise of generating questions from logical forms. We have provided several examples, each comprising an 'Input' logical form and a corresponding 'Subquestion' that we aim to generate. By deconstructing the input logical form into basic components, we can generate questions iteratively until we get the final question. For each 'Subgraph', we can construct a relevant 'Subquestion' phrase to assist in generating the subsequent question in the sequence.*".

## A.4 Effect of Demonstration Order

During the experiment, we made a noteworthy observation regarding the impact of demonstration order on the performance of our method. We conducted a comprehensive exploration of various sorting techniques, including uncertainty-based sorting (Diao et al., 2023), random sorting, and sorting based on the number of logical form jumps. The detailed experimental results are presented in Table 11. It becomes evident that arranging the demonstrations in ascending order of the number of logical form jumps leads to the most favorable outcomes. This finding highlights the structural complexity of logical forms when organizing the demonstrations.

| Method | GQ | | |
|---|---|---|---|
| | B | M | R |
| KQG-CoT+ | 31.24 | 34.94 | 45.36 |
| (a) -Uncertainty | 30.36 | 33.91 | 45.05 |
| (b) -Similarity | 31.20 | 34.63 | 45.28 |
| (c) -Random | 30.81 | 34.26 | 44.91 |
| (d) -L2s | 30.52 | 33.66 | 44.83 |

Table 11: The results of using different sorting methods for demonstrations on the GQ dataset are as follows: Our KQG-CoT+ method is sorted in ascending order of the number of logical form jumps. **Random** sorting is done randomly. **L2S** sorting is performed in ascending order of length. **Uncertainty** sorting is based on descending order of uncertainty values. Lastly, **similarity** sorting is based on descending order of similarity values between the logical forms of demonstrations and tests.

| Method | WQ | | | PQ | | |
|---|---|---|---|---|---|---|
| | B | M | R | B | M | R |
| KQG-CoT+ | 29.73 | 31.08 | 55.14 | 61.71 | 44.27 | 78.41 |
| (a) w/o CoT | 28.75 | 30.12 | 54.24 | 60.83 | 44.06 | 77.88 |
| (b) K-means → Random | 25.02 | 29.37 | 53.16 | 56.42 | 42.61 | 77.03 |
| (c) w/o structure encoding | 28.52 | 29.73 | 54.28 | 60.34 | 43.26 | 77.59 |

Table 12: Ablation study of our KQG-CoT+ method on WQ and PQ.

Demonstrations

Input: (JOIN (R location.country.official_language) (JOIN location.country.languages_spoken romansh language))
Subgraph1: (JOIN location.country.languages_spoken romansh language)
Subgraph2: (JOIN (R location.country.official_language) Subgraph1)
Subquestion1: country languages spoken romansh language
Subquestion2: What is the main language spoken in the country that romansh language is used ?

...

Input: (AND (JOIN people.cause_of_death.parent_cause_of_death drug) (JOIN (R people.deceased_person.cause_of_death) (JOIN film.actor.film (JOIN film.performance.character julia biggs))))
Subgraph1: (JOIN people.cause_of_death.parent_cause_of_death drug)
Subgraph2: (JOIN film.performance.character julia biggs)
Subgraph3: (JOIN film.actor.film Subgraph2)
Subgraph4: (JOIN (R people.deceased_person.cause_of_death) Subgraph3)
Subgraph5: (AND Subgraph1 Subgraph4)
Subquestion1: parent cause of death drug
Subquestion2: performance character julia biggs
Subquestion3: film actor who performance julia biggs
Subquestion4: cause of death of film actor who performance julia biggs
Subquestion5: Which drugs caused the death of the actor who played julia biggs ?

Input: (JOIN (R location.country.currency_used) (JOIN location.country.national_anthem (JOIN government.national_anthem_of_a_country.anthem aruba dushi tera)))

Prediction

Input: (JOIN (R film.performance.actor) (AND (JOIN film.performance.character simon birch) (JOIN film.film.starring (JOIN film.performance.actor ian michael smith))))
Subgraph1: (JOIN film.performance.character simon birch)
Subgraph2: (JOIN film.performance.actor ian michael smith)
Subgraph3: (JOIN film.film.starring Subgraph2)
Subgraph4: (AND Subgraph1 Subgraph3)
Subgraph5: (JOIN (R film.performance.actor) Subgraph4)
Subquestion1: performance character simon birch
Subquestion2: performance actor ian michael smith
Subquestion3: performance actor ian michael smith star in
Subquestion4: performance character simon birch the film that ian michael smith star in
Subquestion5: Who plays simon birch in the movie that ian michael smith acted in ?

Table 13: Prompt with demonstrations and prediction on WQ, the preceding section displays the prompt, and followed section displays the outputs generated by LLMs.

| Demonstrations |
| --- |
| Input: (ARGMIN base.exoplanetology.exoplanet astronomy.astronomical_discovery.discovery_date) |
| Subgraph1: (ARGMIN base.exoplanetology.exoplanet astronomy.astronomical_discovery.discovery_date) |
| Subquestion1: Which exoplanet was first to be found ? |
| |
| ... |
| |
| Input: (AND digicams.digital_camera (AND (lt digicams.digital_camera.weight 250.0^^http://www.w3.org/2001/XMLSchema #float)(JOIN (R digicams.camera_viewfinder_type.digital_cameras) (JOIN digicams.camera_viewfinder_type.digital _cameras Sony Alpha 700)))) |
| Subgraph1: (lt digicams.digital_camera.weight 250.0^^http://www.w3.org/2001/XMLSchema#float) |
| Subgraph2: (JOIN digicams.camera_viewfinder_type.digital_cameras Sony Alpha 700 ) |
| Subgraph3: (JOIN (R digicams.camera_viewfinder_type.digital_cameras) Subgraph2) |
| Subgraph4: (AND Subgraph1 Subgraph3) |
| Subgraph5: (AND digicams.digital_camera Subgraph4) |
| Subquestion1: digital cameras that weight less than 250.0 |
| Subquestion2: viewfinder type digital cameras sony alpha 700 |
| Subquestion3: digital cameras the same viewfinder type as the sony alpha 700 |
| Subquestion4: digital cameras the same viewfinder type as the sony alpha 700 and weight less than 250.0 |
| Subquestion5: Are there any digital cameras that use the same viewfinder as the sony alpha 700 that weight less than 250.0? |
| |
| Input: (AND music.genre (JOIN (R music.genre.parent_genre) (JOIN music.genre.albums confessions tour))) |
| **Prediction** |
| Subgraph1: (JOIN music.genre.albums confessions tour) |
| Subgraph2: (JOIN (R music.genre.parent_genre) Subgraph1) |
| Subgraph3: (AND music.genre Subgraph2) |
| Subquestion1: the music genre albums confessions tour |
| Subquestion2: the albums confessions tour is part of what parent genre |
| Subquestion3: The albums confessions tour is part of what parent genre of a musical genre? |

Table 14: Prompt and prediction on GQ, the preceding section is the prompt, and the blue text following it represents the prediction.

| Demonstrations |
| --- |
| Input: (JOIN (R people.person.gender) (JOIN (R people.person.parents) sviatoslav ii of kiev)) |
| Subgraph1: (JOIN (R people.person.parents) sviatoslav ii of kiev) |
| Subgraph2: (JOIN (R people.person.gender) Subgraph1) |
| Subquestion1: sviatoslav ii of kiev 's parents |
| Subquestion2: What is the gender of sviatoslav ii of kiev 's dad ? |
| |
| ... |
| |
| Input: (JOIN (R people.deceased_person.place_of_death) (JOIN (R people.person.children) (JOIN (R people.person.children) p j kennedy))) |
| Subgraph1: (JOIN (R people.person.children) p j kennedy) |
| Subgraph2: (JOIN (R people.person.children) Subgraph1) |
| Subgraph3: (JOIN (R people.deceased_person.place_of_death) Subgraph2) |
| Subquestion1: p j kennedy 's children |
| Subquestion2: children of p j kennedy 's children |
| Subquestion3: What is the place of death of kid of p j kennedy 's son ? |
| |
| Input: (JOIN (R music.recording.releases) (JOIN (R music.recording.tracks) o holy night)) |
| **Prediction** |
| Subgraph1: (JOIN (R music.recording.tracks) o holy night) |
| Subgraph2: (JOIN (R music.recording.releases) Subgraph1) |
| Subquestion1: o holy night 's tracks |
| Subquestion2: What is the releases of recording of o holy night 's tracks ? |

Table 15: Prompt and prediction on PQ, the preceding section is the prompt, and the blue text following it represents the prediction.