# OpenReview forum: "Prompting Large Language Models with Chain-of-Thought for Few-Shot Knowledge Base Question Generation"
_EMNLP/2023/Conference — EMNLP 2023 Main_

### Official Review · Reviewer_jeur · 2023-08-03

**Soundness:** 4

**Excitement:**

3: Ambivalent: It has merits (e.g., it reports state-of-the-art results, the idea is nice), but there are key weaknesses (e.g., it describes incremental work), and it can significantly benefit from another round of revision. However, I won't object to accepting it if my co-reviewers champion it.

**Missing References:**

Drozdov et al., Compositional Semantic Parsing with Large Language Models. ICLR 2023.

**Paper Topic And Main Contributions:**

The authors propose leveraging Large Language Models (LLMs) and Chain-of-thought (CoT) prompting for the Question Generation over Knowledge Bases (KBQG) tasks. Specifically, the proposed method is broken into 2 steps (1) Supportive Logic Form Selection and (2) Prompt Construction. In the former step, the authors propose encoding the structural knowledge via the logic form. Then, K-Means clustering together with logic form selection are conducted to select the diverse k structures for the sequential prompt construction step. In the next step, motivated by CoT, authors construct reasoning chains from the selected logic form structures as demonstrations for LLM's in-context learning of the KBQG task.

**Questions For The Authors:**

A. For Prompt Construction (Section 3.4), the proposed method seems to adopt the bottom-up approach to extract logic form. How about top-down approach? It would be helpful to compare with other compositional semantic extraction approach of previous works [1].

B. Are the any supportive quantitative metrics and/or visualizations to demonstrate that the sampled k structures from Structure Encoding and Clustering are in fact diverse?

C. How is semantic diversity/ similarity measured (Line 310)?

[1] Drozdov et al., Compositional Semantic Parsing with Large Language Models. ICLR 2023.

**Reasons To Accept:**

1. The work is among the first to explore the capability of LLMs in KBQG tasks.

2. Empirical studies are extensive and provide different aspects of the proposed method.

**Reasons To Reject:**

1. Performance gain of KQG-CoT (presented in Table 2) is marginal over the Auto-CoT, Active-CoT in some metrics, eliciting the questions regarding the absolute necessity of using Logic Form in comparison with narratives in the given task.

2. The proposed work is quite incremental. For instance, K-Means clustering algorithms for k-demonstration selection have been proposed by AutoCoT. The importance of demonstrations order for LLM prompting (in the paper's case, it is the difference between KQG-CoT and vs KQG-CoT+) has also been discussed in previous works.

3. The effectiveness of the proposed Structured Encoding and Clustering in selecting k diverse structures need further quantitative and qualitative evaluation studies.

**Reproducibility:**

3: Could reproduce the results with some difficulty. The settings of parameters are underspecified or subjectively determined; the training/evaluation data are not widely available.

**Reviewer Confidence:**

2: Willing to defend my evaluation, but it is fairly likely that I missed some details, didn't understand some central points, or can't be sure about the novelty of the work.

**Typos Grammar Style And Presentation Improvements:**

Line 17: split
Line 272: drafts -> be drafted
Line 223: consists -> consist

Table 5: Additional explanations/ color coding for the examples of different CoT outputs would be better appreciated.

Table 3,4: Only 1-2 best performing methods for "Full training" would be sufficient since the purpose of the table is to report the previous SoTAs under the full training assumption.

---

> ### Author Rebuttal · Authors · 2023-08-28
>
> We would like to express our sincere gratitude to the reviewers for their thorough and insightful feedback on our manuscript. In the 'Reasons To Reject' and 'Questions For The Authors' sections, you raised some inquiries regarding our work. Below, we will provide a detailed response to the questions you have raised.
>
> **Statement 1:** Performance gain of KQG-CoT (presented in Table 2) is marginal over the Auto-CoT, Active-CoT in some metrics, eliciting the questions regarding the absolute necessity of using Logic Form in comparison with narratives in the given task. \
> **Response:**
> 1) KBQG is an extremely challenging generative task that requires models to comprehend the given logical form and generate coherent, semantically consistent questions. Not only does this demand robust natural language processing capabilities from the models, but it also necessitates a profound understanding of diverse domain knowledge and reasoning abilities. Additionally, the complexity of the few-shot KBQG task goes beyond the technical aspects, encompassing the diversity and openness of answers. The expressions of questions corresponding to logical forms from various domains and topics vary significantly, demanding the models to possess cross-domain and cross-style adaptability.
> 2) As evident from Table 3 and Table 4 in the paper, achieving substantial improvements with new methods is a formidable challenge. Even some well-known efforts to enhance performance in this task have not yielded significant advancements.
> 3) As observed from Table 2, our KQG-CoT approach not only exhibits a value closely approaching that of Active-CoT in one category but also shows evident enhancements in the remaining eight categories compared to Auto-CoT and Active-CoT. Moreover, the performance gain is even more pronounced in KQG-CoT+, with the maximum improvement reaching 2.12%.
> 4) In comparison to the textual format of narratives, the decomposed structured logical form is more suitable for CoT. Structured logical forms have the ability to progressively expand from simple structures to more intricate ones, aligning with the CoT's construction of a series of logically connected thinking steps or ideas. These steps or ideas interconnect with each other, forming a complete thought process.
>
> Taking into account the aforementioned points, it becomes apparent that employing Logic Form is crucial relative to the use of narratives.
>
> **Statement 2:** The proposed work is quite incremental. For instance, K-Means clustering algorithms for k-demonstration selection have been proposed by AutoCoT. The importance of demonstrations order for LLM prompting (in the paper's case, it is the difference between KQG-CoT and vs KQG-CoT+) has also been discussed in previous works. \
> **Response:** We understand the description of "quite incremental" to mean that you may perceive the innovation of our approach as relatively modest compared to existing research. However, while we acknowledge that our method might not be entirely revolutionary in terms of innovation, we believe that our approach represents significant improvements and innovations compared to previous work. This includes several specific aspects:
> 1) While both our method and AutoCoT employ the K-Means method for selecting k-demonstrations, there is a distinction in our approach. Prior to utilizing the K-Means method, we extract the structure of the logical form. Specifically, we cluster the main structure of the logical form using the K-Means method. After clustering, when identifying corresponding logical forms from these structures of the logical forms (where multiple logical forms might share the same structure), we further filter based on semantic similarity. This ensures that the extracted logical forms exhibit diversity. Hence, even though both methods utilize the K-Means method, their emphasis differs. Our method focuses on the extraction of the structure of the logical form and the selection of diverse logical forms based on semantic similarity. **The K-means method is merely one component of our approach; what holds significance is the innovative work we have undertaken to tailor our approach to the demands of KBQG tasks**.
> 2) While previous approaches have discussed the importance of demonstration order for LLM prompting, **there is still no definitive conclusion on the specific optimal order to date**. Published papers, such as those referenced in [2],[3], and [4], have all been exploring viable orders. In our paper, the distinction between the KQG-CoT+ method and the KQG-CoT method lies in our attempt to establish an ordering based on the complexity of the logical form. The experimental results in Table 2 validate the feasibility of this approach for the given task. Therefore, we consider this exploration to be highly meaningful.
> 3) As you mentioned, we are among the first to pioneer the use of LLMs to address KBQG task. Through detailed comparative analysis and experimental results, we have demonstrated that our KQG-CoT+ method offers significant improvements over baselines and traditional approaches. Furthermore, we have conducted ablation experiments to validate the specific contributions of each module in our method. **Our endeavor provides valuable guidance on effectively harnessing the capabilities of LLMs in the realm of KBQG**.
>
> In light of the points discussed above, we believe that characterizing our work as "quite incremental" is not suitable.
>
> **Statement 3:** The effectiveness of the proposed Structured Encoding and Clustering in selecting k diverse structures need further quantitative and qualitative evaluation studies. \
> **Response:** In our paper, we conducted ablation experiments to substantiate the distinct contributions of each module. While you might hold the view that this is not entirely exhaustive and further experimentation is warranted, we promptly executed additional experiments in response. These experiments comprised two main components: firstly, we quantitatively assessed the average semantic similarity between the logical forms extracted using our method and the baseline method at K=8 on the GrailQA dataset; secondly, we employed the visualization tool TSNE to reduce the dimensionality of selected logical form vectors to 2D, thereby generating visual plots. \
> The outcomes of the first segment are presented in the table below, indicating that the logical forms chosen by our method display a lower average semantic similarity. The results of the second segment demonstrate that our method's selected logical forms exhibit greater dispersion, which corroborates the findings of the first part. However, due to constraints in displaying images during the rebuttal phase, we intend to incorporate this information in the revised version of the paper. When considered collectively, these results provide robust support for the efficacy of our proposed method.
> | Method | Average similarity |
> | :---         |     :---:      |
> | Random   |0.285     |
> | Active-CoT    | 0.274       |
> | Auto-CoT    | 0.265       |
> | KQG-CoT    | 0.252       |
>
>
> **Question A:** For Prompt Construction (Section 3.4), the proposed method seems to adopt the bottom-up approach to extract logic form. How about top-down approach? It would be helpful to compare with other compositional semantic extraction approach of previous works [1]. \
> **Response:** Thank you very much for professionally pointing out this new approach to extracting logical forms. It's an important aspect that we overlooked while conducting this work. The reason we chose to adopt a bottom-up approach to extract logical forms is that bottom-up allows for the incremental addition of information, continuously enhancing the complexity of logical forms. This aligns with the CoT concept, which starts with simple questions and gradually addresses more complex ones. In our revised version, we will also explore a top-down method to extract logical forms, thereby enriching the scope of our work.
>
> **Question B:** Are the any supportive quantitative metrics and/or visualizations to demonstrate that the sampled k structures from Structure Encoding and Clustering are in fact diverse? \
> **Response:** This question, along with the preceding Statement 3, pertains to the same aspect, inquiring whether there is concrete data to substantiate the specific impact of sampling k structures from the Structure Encoding and Clustering method. The response aligns with the answer provided for Statement 3.
>
> **Question C:** How is semantic diversity/ similarity measured (Line 310)? \
> **Response:** In lines 310 to 318 of our paper, we elucidate how we measure semantic similarity. The specific details are provided below.
> “Specifically, for the first logical form, we randomly pick up one from the candidates. Then we search logical forms for another structure. We greedily pick up a candidate with least semantic similarity to the selected logical forms, where the similarity is measured by the encoding of the original logical forms. We repeat the process until we have gone through k structures as shown in Figure 2.”
>
> Regarding the section titled "Typos Grammar Style And Presentation Improvements": \
> **1.**“Line 17: split Line 272: drafts -> be drafted Line 223: consists -> consist” \
> **Response:** Thank you very much for your thorough reading and for pointing out the writing errors in our manuscript. These mistakes were due to our oversight, and we apologize for any inconvenience they may have caused to your reading experience. we will carefully review and correct these errors in the upcoming revisions to ensure the paper's accuracy and clarity. \
> **2.**“Table 5: Additional explanations/ color coding for the examples of different CoT outputs would be better appreciated.” \
> **Response:** Thank you very much for your suggestion. In our next version, we will differentiate the examples of different CoT outputs in Table 5 by adjusting font size and color, and by adding additional explanations. \
> **3.**“Table 3,4: Only 1-2 best performing methods for "Full training" would be sufficient since the purpose of the table is to report the previous SoTAs under the full training assumption.” \
> **Response:** In Table 3 and 4, we have provided a detailed presentation of some prominent full-training methods for KBQG developed in recent years. Our intention behind this is twofold: firstly, to demonstrate the level at which the effectiveness of our method resides, and secondly, to offer readers a clear overview of the evolution of KBQG techniques over time.
>
> [1]Drozdov et al., Compositional Semantic Parsing with Large Language Models. ICLR 2023. \
> [2]Lu et al., Fantastically Ordered Prompts and Where to Find Them:Overcoming Few-Shot Prompt Order Sensitivity. \
> [3]Liu et al., What Makes Good In-Context Examples for GPT-3? \
> [4]Diao et al., Active prompting with chain-of-thought for large language models.

---

### Official Review · Reviewer_BZaK · 2023-08-05

**Soundness:** 4

**Excitement:**

3: Ambivalent: It has merits (e.g., it reports state-of-the-art results, the idea is nice), but there are key weaknesses (e.g., it describes incremental work), and it can significantly benefit from another round of revision. However, I won't object to accepting it if my co-reviewers champion it.

**Paper Topic And Main Contributions:**

In the study, the challenge of Question Generation over Knowledge Bases (KBQG) in low-resource scenarios is addressed by introducing a Chain-of-Thought (CoT) inspired prompting technique, reformulating KBQG as a reasoning task. The proposed method, KQG-CoT, selects supportive logical forms from an unlabeled pool and constructs task-specific prompts to guide Large Language Models in generating questions.

**Questions For The Authors:**

Question 1: could you explain how you embed the input logical form (AND medicine.routed_drug (JOIN medcine.routed_drug.marketed_formulatations m.0hqslx))with its structure info (AND r (JOIN r e))? Do you encode the original logical form or just encode its structure info? And why you do not convert the entity ID m.0hqslx into its entity name?

Question 2: If you only encode its structured info, why do you need to cluster them? I think we just need to use string matches to find the most similar structure in the demos. In Figure 2 for a new input, like (COUNT (AND aviation.aircraft_manufacturer (JOIN organization.organization.legal_structure S.A.))), it's better to just select the demo with the same structured info (COUNT (AND r (JOIN r e))), which is the [2] supportive logical forms.

**Reasons To Accept:**

The paper presents a novel approach to the timely challenge of Question Generation over Knowledge Bases in low-resource settings, ingeniously leveraging Chain-of-Thought prompting. The methodological innovation, combined with impressive empirical results surpassing established benchmarks on the PathQuestions dataset, highlights both the study's immediate relevance and its potential to significantly impact the domain. The comprehensive experimentation offers robust evidence of the proposed techniques' effectiveness, making this paper a valuable contribution to the field.

**Reasons To Reject:**

While the paper addresses the relevant challenge of Question Generation over Knowledge Bases, the methodology, inspired by Chain-of-Thought prompting, is not sufficiently differentiated from existing approaches. Additionally, the empirical results, though comprehensive, do not provide a clear justification for the proposed techniques' superiority over established benchmarks.
How can this KBQG help the downstream tasks, like KBQA is not studied? The paper could benefit from deeper analysis and a clearer exposition of the methodological advancements to ensure a more substantial contribution to the field.

**Reproducibility:**

3: Could reproduce the results with some difficulty. The settings of parameters are underspecified or subjectively determined; the training/evaluation data are not widely available.

**Reviewer Confidence:**

4: Quite sure. I tried to check the important points carefully. It's unlikely, though conceivable, that I missed something that should affect my ratings.

**Typos Grammar Style And Presentation Improvements:**

Figure 2. In Prompt Construction Subgraph1: (JSON sports.sport.team_coaches John Russo) should be (JOIN).

---

> ### Author Rebuttal · Authors · 2023-08-28
>
> We wholeheartedly express our gratitude to you for offering comprehensive and insightful feedback on our manuscript. We are both honored and pleasantly surprised by your high evaluation. However, in the same vein, you have also highlighted the shortcomings of our work in the "Reasons To Reject" section and raised two questions. Below, we will provide detailed explanations for both aspects.
>
> **Statement 1:** While the paper addresses the relevant challenge of Question Generation over Knowledge Bases, the methodology, inspired by Chain-of-Thought prompting, is not sufficiently differentiated from existing approaches.\
> **Response:** As you mentioned in the "Reasons To Accept" section, our approach is innovative and adeptly employs CoT prompts to address KBQG challenges in low-resource environments. As showcased in our paper's Methodology (Section 3) and Figure 2, the innovation within our method is primarily evident through the following facets:
> 1. As we strive to identify supportive logical forms, we meticulously consider the inherent structures of these forms. This allows us to extract supportive logical forms that exhibit semantic diversity.
> 2. In the utilization of the CoT methodology, we astutely leverage the structured nature of logical forms. We progressively expand these forms from simpler, shorter constructs to more intricate and elaborate structures, seamlessly aligning them with the CoT framework.
> 3. Importantly, we are pioneering the integration of LLMs and CoT to tackle the KBQG task. Additionally, we arrange demonstrations based on the complexity of the logical forms. The experimental results presented in Table 2 serve to validate the feasibility of this approach for the given task. Consequently, we firmly believe that this exploration bears significant significance not only within KBQG tasks but also in other tasks necessitating structured inputs.
>
> In summary, our approach not only draws insights from previous methods but also introduces innovation, creating a notable distinction from existing approaches.
>
> **Statement 2:** Additionally, the empirical results, though comprehensive, do not provide a clear justification for the proposed techniques' superiority over established benchmarks.  \
> **Response:** Prior research [1] has demonstrated that the performance of In-context learning (ICL) (of which CoT is a subset) heavily relies on **Demonstration Selection**, **Order of Demonstration Examples**, and **Reasoning Steps Formatting**. Therefore, the design of our approach takes these three aspects into careful consideration. Furthermore, in the "More Analysis" (Section 4.5) section of our paper, we conduct **Ablation Study** to validate the impact of each component. From the results presented in Tables 7, 8, and 9, the contributions of each part to the overall effectiveness of our method are evident. Thus, both theoretically (an aspect that does not require further validation in our work again) and experimentally, there are clear reasons to support the superiority of our proposed method over established benchmarks.
>
> **Statement 3:** How can this KBQG help the downstream tasks, like KBQA is not studied? \
> **Response:** In the "Introduction" section, we mentioned that KBQG can assist QA systems in data augmentation and aid in generating coherent questions for dialogue systems, among other applications. As a result, it has been garnering increasing interest from both industry and academia. Past research[2],[3] has already demonstrated that for solving KBQA and other tasks, effective data augmentation through KBQG can enhance training data and the quality of generated questions has a significant impact on the performance improvement of the task models. \
>  Our proposed method, requiring only the extraction of a few logical forms from the knowledge graph, can generate high-quality questions. This feature positions it as a valuable asset for downstream KBQA tasks. The specific benefits of KBQG for various downstream tasks and the mechanisms behind these benefits lie outside the scope of this study and are part of our future research plans. We intend to utilize the KBQG technique developed in this study for data augmentation in addressing KBQA tasks under few-shot settings, as well as quantifying the impact of increased data volume on KBQA improvement in our future research endeavors.
>
> **Question 1:** could you explain how you embed the input logical form (AND medicine.routed_drug (JOIN medcine.routed_drug.marketed_formulatations m.0hqslx))with its structure info (AND r (JOIN r e))? Do you encode the original logical form or just encode its structure info? And why you do not convert the entity ID m.0hqslx into its entity name? \
> **Response:** We perform embedding on the structures of logical forms because the extracted structures of logical forms does not include entity IDs, so eliminating the need for conversion to their corresponding entity names. \
> Initially, we attempted to embed the original logical forms and directly apply k-means clustering. However, we found that this approach yielded suboptimal results. We find that the diversity in logical forms, stemming from their inclusion of keywords, relations, and mid-components, might lead to this outcome. Considering that the structures of logical forms share similarities with the arrangement of sentence parsing syntax trees, we chose to extract and incorporate the principal structural components of these logical forms. So, we perform embedding on the structure of the logical form and then proceed with clustering.
>
> **Question 2:** If you only encode its structured info, why do you need to cluster them? I think we just need to use string matches to find the most similar structure in the demos. In Figure 2 for a new input, like (COUNT (AND aviation.aircraft_manufacturer (JOIN organization.organization.
> legal_structure S.A.))), it's better to just select the demo with the same structured info (COUNT (AND r (JOIN r e))), which is the [2] supportive logical forms. \
> **Response:** According to intuition, it might be more effective to select K most similar structures and construct demonstrations from them. However, we chose not to pursue this approach due to several reasons:
> 1. Based on findings from the auto-CoT paper, using a method that selects the most similar structures and then combining automated methods for constructing CoT reasoning steps can lead to errors and subpar results.
> 2. We are addressing the issue of few-shot KBQG, which involves manually constructing intermediate reasoning steps when forming CoT-style demonstrations. If we were to construct demonstrations anew for each selected logical form, it would be cumbersome, time-consuming, and resource-intensive.
> 3. Furthermore, the method you mentioned, which we experimented with under the "Similarity" row in Table 8 of Appendix A and he experimental results demonstrate that our proposed method significantly outperforms the approach you mentioned.
>
> Hence, our method involves initially using clustering to select representative logical forms, followed by manual construction of intermediate reasoning steps. Regardless of the input, the demonstrations remain consistent. This approach ensures that even if we need to manually create intermediate reasoning steps for K demonstrations, it remains manageable and guarantees the correctness of the reasoning process. In conclusion, both theoretically and experimentally, it has been demonstrated that the find the most similar structure of the input approach is not feasible.
>
> **Regarding your observation on Figure 2, where you pointed out "In Prompt Construction Subgraph1: (JSON sports.sport.team_coaches John Russo) should be (JOIN),"** we want to express my heartfelt gratitude for your careful reading. This writing error was a result of my oversight. We will thoroughly review and rectify these errors in the subsequent revisions to ensure the accuracy and clarity of the paper.
>
> [1]Dong et al., A Survey for In-context Learning. \
> [2]Guo et al., DSM: Question Generation over Knowledge Base via Modeling Diverse Subgraphs with Meta-learner. \
> [3]Maharana et al., GRADA: Graph Generative Data Augmentation for Commonsense Reasonin

---

### Official Review · Reviewer_PKjA · 2023-08-12

**Soundness:** 4

**Excitement:**

4: Strong: This paper deepens the understanding of some phenomenon or lowers the barriers to an existing research direction.

**Paper Topic And Main Contributions:**

The paper utilized LLMs with few-shot learning to tackle the task of Question Generation over Knowledge Bases (KBQG) with the Chain of Thought (CoT) approach.

**Questions For The Authors:**

Question A: What are the primary contributions of this paper in comparison to Auto-CoT?
Question B: Is the order of the selected logical forms significant? What is the reasoning behind choosing the last subquestion as the final prediction?


**Reasons To Accept:**

The contribution of this work is that it leverages Language Models (LLMs) with few-shot learning and employs the Chain of Thought (CoT) approach for Question Generation over Knowledge Bases (KBQG).

**Reasons To Reject:**

It is important to clarify the specific contributions of this paper in relation to Auto-CoT (Zhang et al., 2023c).

**Reproducibility:**

4: Could mostly reproduce the results, but there may be some variation because of sample variance or minor variations in their interpretation of the protocol or method.

**Reviewer Confidence:**

4: Quite sure. I tried to check the important points carefully. It's unlikely, though conceivable, that I missed something that should affect my ratings.

---

> ### Author Rebuttal · Authors · 2023-08-28
>
> We sincerely thank you for your meticulous and insightful feedback on our manuscript. In response to your feedback, we are providing a general response to the questions you've raised here.
>
> **Question A:** What are the primary contributions of this paper in comparison to Auto-CoT?\
> **Response :** The primary contributions of this paper in comparison to Auto-CoT lie in the following ways:
> 1. While our method and auto-CoT utilize K-means clustering to select supportive examples for constructing demonstrations, we take a different approach regarding the logical forms from input questions. We holistically consider the structural information of logical forms. Illustrated in the Supportive Logical Forms Selection (Section 3.3) of our paper, our method initially extracts structures of logical forms, which omit the semantics of the inputs at this stage. Subsequently, we cluster and sample the K main structures of logical forms. By amalgamating the sampled structures of logical forms with semantic similarity, we employ a greedy approach to select logical forms with the lowest average semantic similarity. As evident from our approach, the manner in which we unravel the semantic and syntactic components for logic forms differs from that of Auto-CoT, yet it holds significant importance for KBQG tasks.
> 2. Our utilization of the CoT methodology varies. Unlike Auto-CoT, where the output is generated with single turn, the generation process of our method follows incremental expansion. Progressing from shorter to more extensive structures, structured inputs can be incrementally transitioned, thereby aligning more closely with the CoT concept. In our paper, for input logical forms, we progressively expand them using their keywords. This ensures that each guiding segment of CoT remains accurate a feat not easily achievable with unstructured inputs. Moreover, to our best known, we are the pioneers in employing LLMs and CoT to tackle the KBQG task.
> 3. Of utmost significance, prior research [1], [2] underscores the crucial importance of the order of demonstrations. This aspect is overlooked by auto-CoT, whereas our method arranges demonstrations based on the complexity of corresponding logical forms. Our results indicate superiority over random ordering. We order demonstrations based on the complexity of the logical form. The experimental results in Table 2 validate the feasibility of this approach for the given task. Therefore, we believe that this exploration holds great significance in KBQG tasks or other tasks involving structured inputs.
> In summary, although our approach shares some similarities with auto-CoT, our primary emphasis lies in the novel contributions our method brings to addressing the KBQG task.
>
>
> **Question B:** Is the order of the selected logical forms significant? What is the reasoning behind choosing the last subquestion as the final prediction?\
> **Response:** Just as we mentioned in our paper, the logical forms we have selected will subsequently form the demonstrations in the prompt. Both [1] and [2] emphasize the importance of the order of demonstrations. Moreover, KBQG tasks aim at generating questions with complex expression and nested logics. Hence, a progressive generation order is important, which has been proven by comparing the KQG-CoT and KQG-CoT+ methods in Table 2 of our paper. So the order of the selected logical forms is very significant.
> Regarding the decision to choose the last subquestion as the final prediction, this rationale becomes evident through both the Prompt Construction (Section 3.4) outlined in the paper and Figure 2. Each substructure of the logical form corresponds to a distinct phrase, and as we iteratively elaborate on these substructures, the corresponding phrases naturally grow in comprehensiveness. As we approach the culmination of the logical form in its entirety, the corresponding statement reaches its fullness, ultimately aligning with the answer we aim to achieve.
>
> [1] Lu et al.,  Fantastically Ordered Prompts and Where to Find Them:Overcoming Few-Shot Prompt Order Sensitivity.\
> [2] Dong et al.,  A Survey for In-context Learning.

---

### Meta-Review · Area_Chair_2uzH · 2023-09-18

**Recommendation:** 4

**Metareview:**

This paper proposes a chain of thought prompting-based approach for question generation over KBs in low-resource settings. The reviewers all believe the task is important, and it has been under-explored with LLMs in prior work. On the negative side, the improvements of the proposed method over competitors (e.g., Auto-CoT) are fairly small, which does call into question the method's generalizability. I would suggest that the authors consider adding a more thorough and well-documented human evaluation to the next version of their paper, as that would help determine its effectiveness more than simple (and flawed) string overlap-based methods. The authors were forthcoming with new results and details during the discussion period, which is much appreciated by both the reviewers and this AC. Overall, there are flaws with this paper, but I believe it will  still make a valuable contribution to EMNLP (either main conference or findings).

---

### Decision · Program_Chairs · 2023-10-07

**Decision:**

Accept-Main

**Comment:**

This paper proposes a chain of thought prompting-based approach for question generation over KBs in low-resource settings. The reviewers all believe the task is important, and it has been under-explored with LLMs in prior work. On the negative side, the improvements of the proposed method over competitors (e.g., Auto-CoT) are fairly small, which does call into question the method's generalizability. I would suggest that the authors consider adding a more thorough and well-documented human evaluation to the next version of their paper, as that would help determine its effectiveness more than simple (and flawed) string overlap-based methods. The authors were forthcoming with new results and details during the discussion period, which is much appreciated by both the reviewers and this AC. Overall, there are flaws with this paper, but I believe it will  still make a valuable contribution to EMNLP (either main conference or findings).